# Quantitative detection and prognostic value of antibodies against M-type phospholipase A2 receptor and its cysteine-rich ricin domain and C-type lectin domains 1 and 6-7-8 in patients with idiopathic membranous nephropathy

**Xiaobin Liu[1]©, Jing Xue[1]©, Ting Li[2]©, Qingqing Wu[2], Huiming Sheng[3], Xue Yang[4], Bo Lin[5], Xiumei Zhou[2], Yuan Qin[2], Zijian Huang[6], Leting Zhou[1], Liang Wang[1]\*, Zhigang Hu[1,4]\*, Biao Huang[2]\***

1 Wuxi Medical Center, Department of Nephrology, The Affiliated Wuxi People's Hospital of Nanjing Medical University, Nanjing Medical University, Wuxi, China, 2 College of Life Sciences and Medicine, Zhejiang Sci-Tech University, Hangzhou, China, 3 Tong Ren Hospital, Shanghai Jiao Tong University School of Medicine, Shanghai, China, 4 Wuxi Children's Hospital, Wuxi, China, 5 Zhejiang Provincial People's Hospital, Hangzhou, China, 6 School of Life Science, Westlake University, Hangzhou, China

© These authors contributed equally to this work.
\* wangliang_wuxi@126.com (LW); jswxhb@163.com (BH); jswxhzg@163.com (ZH)

## Abstract

### Background

M-type phospholipase A2 receptor (PLA2R) is the major autoantigen in adult idiopathic membranous nephropathy (IMN). Although reactive epitopes in the PLA2R domains have been identified, the clinical value of these domains recognized by anti-PLA2R antibodies remains controversial. Accordingly, this study aimed to quantitatively detect changes in the concentrations of different antibodies against epitopes of PLA2R in patients with IMN before and after treatment to evaluate the clinical value of epitope spreading.

### Methods

Highly sensitive time-resolved fluorescence immunoassay was used to quantitatively analyze the concentrations of specific IgG and IgG4 antibodies against PLA2R and its epitopes (CysR, CTLD1, CTLD6-7-8) in a cohort of 25 patients with PLA2R-associated membranous nephropathy (13 and 12 in the remission and non-remission groups, respectively) before and after treatment, and the results were analyzed in conjunction with clinical biochemical indicators.

### Results

The concentration of specific IgG (IgG4) antibodies against PLA2R and its epitopes (CysR, CTLD1 and CTLD6-7-8) in non-remission group was higher than that in remission group.

**Data Availability Statement:** All relevant data are within the manuscript and its Supporting Information files.

**Funding:** This study was supported by the Social Development Fund of Zhejiang Province [No. LGF20H200008]; Chinese National Natural Science Foundation [No. 82172336,81672083,82070730]; Key Research and Development Project of Zhejiang Province (No. 2022C03118), the Natural Science Foundation of Zhejiang Province(LQ23H050005), Key Research and Development Project of Hangzhou (No. 202004A23),Precision medicine key project of Wuxi Health Commission [No. J202001], theTop Talent Support Program for young and middle-aged people of Wuxi Health Committee [HB2020008],Wuxi Medical Innovation Team Project [CXTD2021010],Specialized Disease Cohort of Wuxi Medical Center of Nanjing Medical University(WMCC202316). The funders had no role in study design, data collection and analysis, decision to publish, or preparation of the manuscript.

**Competing interests:** The authors have declared that no competing interests exist.

The multipliers of elevation of IgG (IgG4) antibody were 5.6(6.2) fold, 3.0(24.3) fold, 1.6(9.0) fold, and 4.2(2.6) fold in the non-remission/remission group, respectively. However, the difference in antibody concentrations between the two groups at the end of follow-up was 5.6 (85.2), 1.7 (13.1), 1.0 (5.1), and 1.5 (22.3) times higher, respectively. When detecting concentrations of specific IgG antibodies against PLA2R and its different epitopes, the remission rate was 66.67% for only one epitope at M0 and 36.36% for three epitopes at M0. When detecting concentrations of specific IgG4 antibodies against PLA2R and its different epitopes, the remission rate was 100.00% for only one epitope at M0 and 50.00% for three epitopes at M0. A trivariate logistic regression model for the combined detection of eGFR, anti-CTLD678 IgG4, and urinary protein had an AUC of 100.00%.

## Conclusion

Low concentrations of anti-CysR-IgG4, anti-CTLD1-IgG4, and anti-CTLD6-7-8-IgG4 at initial diagnosis predict rapid remission after treatment. The use of specific IgG4 against PLA2R and its different epitopes combined with eGFR and urinary protein provides a better assessment of the prognostic outcome of IMN.

## 1. Introduction

Membranous nephropathy (MN) is an immune-mediated glomerular disease-causing nephrotic syndrome in adults. The pathological characteristics of MN include pathological changes in the glomerular base, with a large amount of immune complexes deposited on the epithelial side of the basement membrane of the glomerular capillary [1]. Clinically, MN generally manifests as proteinuria, hypoproteinemia, severe edema, and hyperlipidemia, which are the primary manifestations of nephrotic syndrome. Depending on its cause, adult MN can be divided into secondary membranous nephropathy (SMN) and idiopathic membranous nephropathy (IMN). SMN is caused by systemic diseases, such as systemic lupus erythematosus, rheumatoid arthritis, hepatitis B virus infection, and tumors. IMN is an autoimmune disease. In 2009, Beck et al. [2] reported that M-type phospholipase A2 receptor (PLA2R) is the principal target antigen in 70%–80% of IMN cases. Podocyturia, the mark of podocyte injury, was found to correlate with PLA2R antibody levels and prognosis in membranous nephropathy [3]. The deposition complex of IgG-positive grain sediments, in which the main component of the immune complex is IgG4, a subclass of IgG bound to PLA2R, can be detected using immunohistochemistry and electron microscopy [4]. PLA2R belonging to the mannose receptor family is divided into extracellular, transmembrane, and intracellular parts with a relative molecular weight of 180 kDa. The extracellular part is composed of a cysteine-rich ricin domain (CysR), a fibronectin type II domain (FN II), and eight tandem C-type lectin domains (CTLDs) from the N-terminal to the C-terminal [5].

The recognition anti-PLA2R antibody epitopes is conformationally dependent, and IMN may thus also be a molecular conformational autoimmune disease. Seitz-Polski [6] proposed CysR, CTLD1, and CTLD7 as the dominant epitopes in PLA2R; that is, their original conformations are easily changed by the renal microenvironment, resulting in the expansion of PLA2R epitopes, which are then recognized by the immune system. This phenomenon allows anti-PLA2R antibodies to target at least three epitopic domains of PLA2R. Seitz-Polski et al. suggested that an analysis of PLA2R epitopes by characterization and spreading is a powerful

tool for monitoring disease severity and stratifying patients by renal prognosis (Fig 1). Subsequently, Reinhard et al. [7] found that CTLD8 is also an epitope of PLA2R. Antibodies recognizing at least two epitopes were present in patients with IMN at the first visit and most patients were in remission after treatment, there was little difference in the presence or absence of epitope migration, which was inconsistent with Seitz-Polski's findings. It is thus unclear whether or not the prognosis of a disease can be predicted using epitope spreading remains controversial. In addition, most of the above studies used qualitative-based western blot analysis, and the sensitivity of the assay was limited. Therefore, to evaluate the clinical value of epitope spreading in this study, we used highly sensitive time-resolved fluorescence immunoassay (TRFIA) and aimed to quantitatively detect changes in the concentrations of different antibodies against epitopes of PLA2R in patients with IMN before and after treatment. Considering that CTLD7 alone has no immunoactivity, we used CTLD6-7-8 as the C-terminal epitopes of PLA2R to detect specific antibodies in this study.

## 2. Materials and methods

### 2.1 Chemicals and instrumentation

PLA2R-IgG, CysR-IgG, CTLD1-IgG, CTLD6-7-8-IgG, PLA2R-IgG4, CysR-IgG4, CTLD1-IgG4, and CTLD6-7-8-IgG4 TRFIA kits (including PLA2R, CysR, CTLD1, CTLD6-7-8 antigen, $Eu^{3+}$-labeled mouse anti-human IgG4 antibody, $Eu^{3+}$-labeled goat anti-human IgG antibody, different concentrations of anti-PLA2R-IgG reference standards, coating buffer, blocking buffer, assay buffer, enhancement solution, and washing solution) were purchased from Zhejiang Boshi Biotechnology Co., Ltd.(Zhejiang, China). The 96-well plates were purchased from Xiamen Yunpeng Technology Development Co., Ltd. (Fujian, China). A micro-oscillator was purchased from Jiangsu Kangjian Medical Products Co., Ltd. (Jiangsu, China). A Barnstead water purification system was used to produce pure water. The DR6608 time-resolved fluorescence immunoassay analyzer was purchased from Foshan Da'an Medical Equipment Co., Ltd. (Guangdong, China).

### 2.2 Sera and patients

From March 2016 to December 2016, we performed a cohort studyof 25 patients with IMN at Wuxi People's Hospital, which is affiliated with Nanjing Medical University. All patients were diagnosed with MN by renal biopsy. Secondary kidney diseases such as tumors, hepatitis, and antoimmune diseases were excluded. The patients underwent treatment with immunosuppressants (including rituximab, tacrolimus, cyclosporin, and cyclolinamide) based on the maximum tolerated dose of RASI. The immunosuppressant selection was based on a combination of risk stratification according to KDIGO guidelines and patient wishes. After 6 months of treatment, the patients were divided into remission and non-remission groups based on clinical indications. Twenty five serum samples were collected at the time of renal biopsy and 6-month follow-up. Twenty serum samples from healthy volunteers were used as normal controls. The study protocol was approved by the Institutional Review Board of the Affiliated Wuxi People's Hospital of Nanjing Medical University (KYL2016001) and conducted in accordance with the ethical principles stated in the Declaration of Helsinki. Informed consent was obtained from all participants.

### 2.3 PLA2R-, CysR-, CTLD1-, and CTLD6-7-8-specific TRFIAs [8]

**2.3.1 Antigen coating.** Purified PLA2R or domain proteins (CysR, CTLD1, or CTLD6-7-8) were coated in 96-well plates and then diluted with a coating buffer solution to 2 µg/mL.

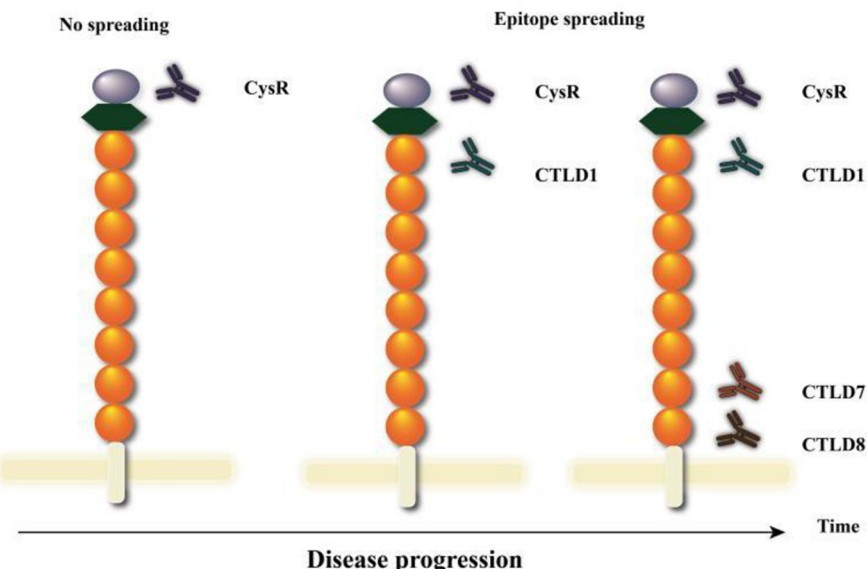

**Fig 1. Schematic of epitope spreading of PLA2R.**

The diluted antigens (100 μL) were added to each well of the 96-well plates, which were incubated overnight at 4 C. The next day, the plates were washed once with washing solution, and each well was blocked by adding 150 μL of blocking buffer for 2 h at room temperature. After blocking, the plates were dried and stored at -20˚C [9].

**2.3.2 TRFIA for IgG-specific antibodies.** The serum samples were diluted 1:100 in assay buffer and added (100 μL/well) in quadruplicate to the antigen-coated plates. After 1 h of incubation at 25˚C in a shaker, the plates were washed thrice with a wash buffer, added with $Eu^{3+}$-labeled goat anti-human IgG antibody (diluted 1:100 in assay buffer) at 100 μL/well, and then incubated for 1 h at 25˚C in a shaker. After six washes, the enhancement solution was added (200 μL/well), and the reaction was let to continue for 5 min at 25˚C in a shaker [10]. Fluorescence was obtained using a DR6608 time-resolved fluorescence immunoassay analyzer (Da'an Gene, Guangzhou, China).

**2.3.3 TRFIA for IgG4-specific antibodies.** Serum samples were diluted 1:20 and added (100 μL/well) in quadruplicate to antigen-coated plates. After 1 h of incubation at 25˚C in a shaker, the plates were washed thrice with a wash buffer. In the second incubation, $Eu^{3+}$-labeled mouse anti-human IgG4 antibodies (1:100 dilution) were used [11], and the remaining procedures were the same as those performed in the TRFIA for IgG-specific antibodies.

**2.3.4 Definitions and calculations.** Twenty serum samples from healthy volunteers were used to define the normal range, with mean + 2 SD as the cut-off value. Values > mean + 2 SD were considered positive.

Remission was defined based on the guidelines of Kidney Disease Improving Global Outcomes. Partial remission (PR) was defined as urinary protein < 3.5 g/24 h with a reduction of at least 50% and a stable renal function. Complete remission (CR) was defined as urinary protein <0.3 g/24 h with a normal or stable glomerular filtration rate. No response (NR) was considered when a <50% reduction in proteinuria or worsening of serum creatinine was observed.

### 2.4 Statistical analyses

For descriptive statistics, normally distributed variables are presented as the mean ± SD, non-normally distributed variables as median (interquartile range), and categorical variables as percentage. The Shapiro–Wilk test was used to assess whether or not a variable had a normal distribution. Qualitative variables were compared using the chi-square test or Fisher's exact test, whereas quantitative variables were compared using Student's t-test or Wilcoxon–Mann–Whitney test. Multiple comparisons were performed using one-way analysis of variance or Kruskal–Wallis test, depending on whether or not the data were normally distributed. Pearson's correlation coefficient was used to evaluate correlations. Multivariable logistic regression analysis was used to determine covariates associated with remission. All statistical data were analyzed using GraphPad Prism software (version 8.0) and SPSS version 26.0. All $P$ values were two-tailed, and the results were considered significant at $P < 0.05$.

## 3. Results

Twenty-five patients with IMN were evaluated and followed up for six months, during which time they were treated with immunosuppressive therapy. The patients were divided into a remission group (including PR and CR, n = 13) and a non-remission group (n = 12) depending on their clinical conditions at the last follow-up. Pathological biopsies at initial diagnosis showed minimal differences between the groups. Age, sex, hypertension, and triglyceride levels at initial diagnosis did not significantly differences between the two groups; however, significant differences were noted in proteinuria, serum albumin, creatinine, and total cholesterol levels and estimated glomerular filtration rate at initial diagnosis. Only anti-PLA2R-IgG and anti-CysR -IgG were significantly different between the remission and non-remission groups at M0, while concentrations of specific IgG4 targeting PLA2R and all its epitopes were significantly different in both the remission and non-remission groups at M0. ($P < 0.05$, Table 1, Figs 2 and 3).

Normal values are expressed as the mean ± SD; non-normal values are median (interquartile range); qualitative values are numbers (%). "Re" represents the remission group and "No-Re" represents the non-remission group.

To evaluate the test performances, the ROC curve was analyzed based on the serum anti-PLA2R-IgG and anti-PLA2R-IgG4 levels at initial diagnosis. The AUC of anti-PLA2R-IgG4 and anti-PLA2R-IgG was 0.8320 versus 0.8352, respectively. We further used multiple logistic regression to analyze the factors affecting the remission of urinary protein. Sixteen factors (including age, gender, hypertension, proteinuria, serum albumin, serum creatinine, serum total cholesterol, triglyceride, eGFR, anti-PLA2R-IgG, anti-PLA2R-IgG4, anti-CysR-IgG, anti-CysR-IgG4, anti-CTLD1-IgG, anti-CTLD1-IgG4, anti-CTLD678-IgG, and anti-CTLD678-IgG4) in the multivariate analysis. The model was screened using the forward method based on the Akaike information criterion(AIC); it included eGFR, CTLD678-IgG4, and urine protein. Trivariate logistic regression models combining eGFR, anti-CTLD678 IgG4, and urinary protein were found to have an AUC reaching 100% (Fig 4).

With respect to analyzing the number of epitopes, the number of epitopes targeted (1 vs 2 vs 3) was from (4 vs 5 vs 4) turn into (9 vs 3 vs 1) at M0 and M6 in the remission groups, and from (2 vs 3 vs 7) turn into (7vs 4 vs 1) at M0 and M6 in the non-remission groups, respectively, with respect to the detection of IgG antibodies using epitopes. The proportion of three epitopes was significantly higher in the non-remission group, the remission rate was 66.67% for only one epitope at M0 and 36.36% for three epitopes at M0 when concentrations of specific IgG antibodies against PLA2R different epitopes were detected. When concentrations of specific IgG4 antibodies against PLA2R and its different epitopes were detected, the remission

**Table 1. Clinical characteristics, anti-PLA2R antibody concentrations, and domain-specific antibody concentrations at initial diagnosis and 6 months follow-up (n = 25).**

| clinical characteristics | initial diagnosis(M0) | | | | follow-up(M6) | | | |
|---|---|---|---|---|---|---|---|---|
| | remission group ($n = 13$) | non-remission group ($n = 12$) | P Value | No-Re/ Re | remission group ($n = 13$) | non-remission group ($n = 12$) | P Value | No-Re/ Re |
| Age (years) | 49 ± 14 | 54 ± 8 | 0.24 | | | | | |
| Sex (male/female) | 6/7 | 9/3 | 0.23 | | | | | |
| RASI | 13(100%) | 12(100%) | | | | | | |
| Immunosuppressive therapy: CTX/ CNI/RTX | 12/1/0 | 11/0/1 | | | | | | |
| Hypertension[numbers (%)] | 5 (38.5%) | 5 (41.7%) | 0.60 | | 2 (15.4%) | 3 (25%) | 0.65 | |
| Proteinuria (g/d) | 5.6 ± 0.9 | 7.7 ± 1.6 | 0.001 | | 0.3 (0.1–1.8) | 5.5 (4.2–7.6) | <0.001 | |
| Serum albumin (g/L) | 23.4 ± 4.5 | 17.5 ± 4.2 | 0.003 | | 33.4 (32.6–36.7) | 20.9 (16.5–26.1) | <0.001 | |
| Serum creatinine (μM) | 66.0(58.3–76.3) | 104.7(77.6–127.0) | 0.001 | | 72.8 (63.2–80.5) | 105.0(78.4–160.6) | 0.01 | |
| Serum total cholesterol (mM) | 6.9 ± 1.6 | 8.7 ± 2.2 | 0.03 | | 4.7 ± 1.0 | 6.6 ± 1.0 | <0.001 | |
| Triglyceride (mM) | 2.4 (1.7–3.5) | 2.2 (1.6–3.1) | 0.57 | | 1.7 ± 1.0 | 2.3 ± 0.8 | 0.13 | |
| eGFR (mL/min/1.73m$^2$) | 104.2 ± 12.1 | 68.2 ± 24.4 | <0.001 | | 94.9 ± 17.8 | 67.6 ± 33.2 | 0.02 | |
| Anti-PLA2R-IgG (RU/mL) | 19.9(11.2–59.4) | 111.1(60.9–160.0) | 0.001 | 5.6 | 6.4 (4.8–8.8) | 36.0(15.0–48.1) | 0.001 | 5.6 |
| Anti-PLA2R-IgG4 (RU/mL) | 119.6(18.6–525.8) | 742.7(420.1–1126.7) | 0.004 | 6.2 | 2.4 (0.8–3.9) | 204.4(82.0–428.7) | <0.001 | 85.2 |
| Anti-CysR-IgG (RU/mL) | 45.9(24.5–72.0) | 139.0(66.0–582.0) | 0.003 | 3.0 | 17.1 (12.1–25.3) | 29.3(15.8–56.7) | 0.14 | 1.7 |
| Anti-CysR-IgG4 (RU/mL) | 36.3(12.2–421.9) | 880.8(467.2–1813.1) | 0.002 | 24.3 | 5.7 (4.7–6.8) | 74.9 (33.2–330.5) | <0.001 | 13.1 |
| Anti-CTLD1-IgG (RU/mL) | 77.7 (41.9–122.7) | 125.3 (54.9–268.3) | 0.10 | 1.6 | 60.0 (34.4–76.0) | 60.3 (33.0–92.2) | 0.91 | 1.0 |
| Anti-CTLD1-IgG4 (RU/mL) | 6.6 (4.0–15.6) | 59.2 (14.1–931.8) | 0.004 | 9.0 | 3.1 (2.3–4.6) | 15.8 (4.2–143.5) | 0.002 | 5.1 |
| Anti-CTLD6-7-8-IgG (RU/mL) | 42.4 (23.8–93.7) | 177.0 (29.7–585.9) | 0.06 | 4.2 | 14.8 (12.1–22.1) | 22.6 (8.5–25.5) | 0.55 | 1.5 |
| Anti-CTLD6-7-8-IgG4 (RU/mL) | 154.3 (23.4–451.0) | 406.9 (199.7–601.2) | 0.03 | 2.6 | 1.3 (0.7–5.2) | 29.0 (7.2–216.2) | 0.001 | 22.3 |

GFR (CKD-EPI) = a × (serum creatinine/b)c × (0.993) age

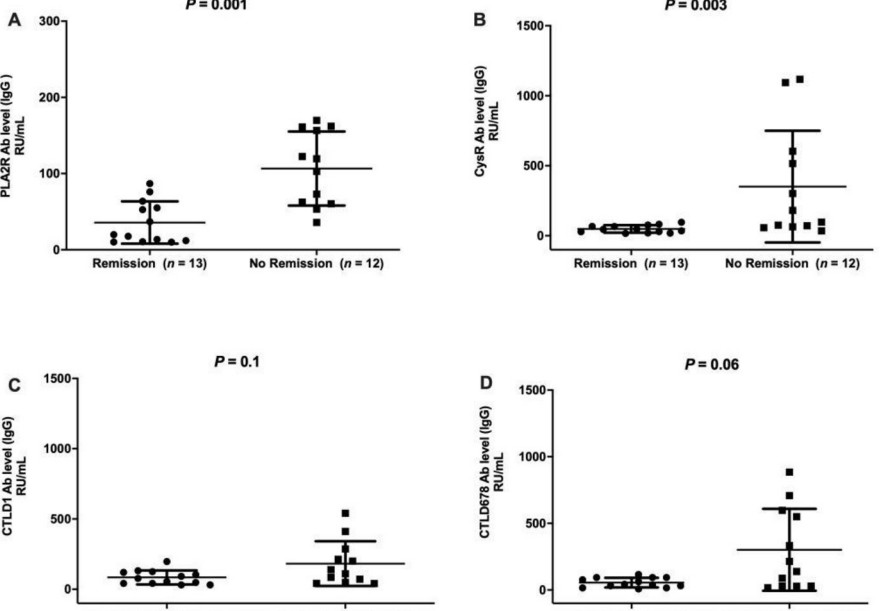

**Fig 2. Concentrations of IgG antibodies against PLA2R and its CysR, CTLD1, and CTLD6-7-8 epitopes in patients from the remission and non-remission groups at initial diagnosis.**

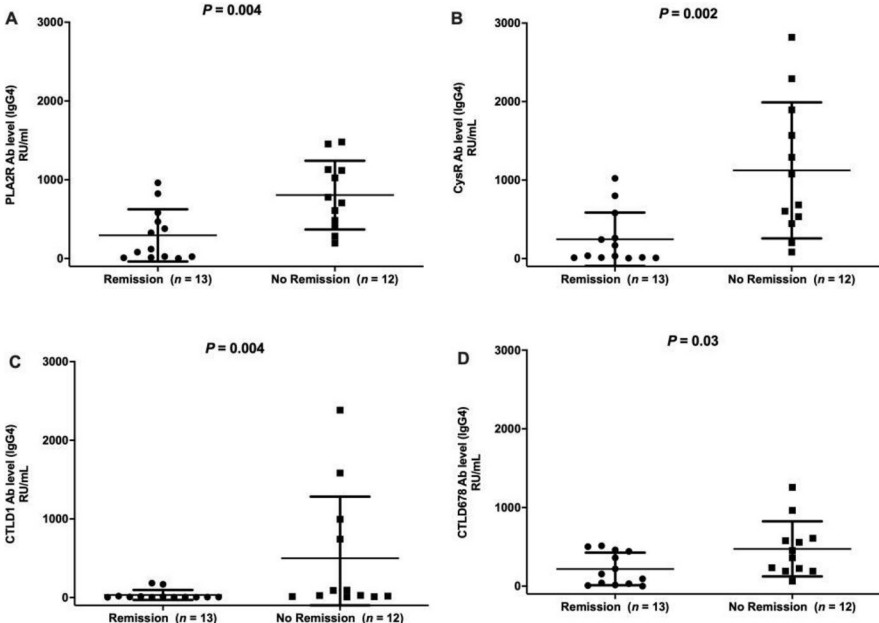

**Fig 3. Concentrations of IgG4 antibodies against PLA2R and its CysR, CTLD1, and CTLD6-7-8 epitopes in patients from the remission and non-remission groups at initial diagnosis.**

rate was 100% for only one epitope at M0 and 50.00% for three epitopes at M0. When concentrations of specific IgG4 antibodies against PLA2R different epitopes were detected, the remission rate was 100% for only one epitope at M0 and 50.00% for three epitopes at M0 (Fig 5). A further quantitative analysis of the antibody concentration was conducted, and anti-PLA2R-IgG and the anti-CysR-IgG antibody concentrations were found to be significantly higher in the patients from the non-remission group than in the remission group at initial diagnosis. The median concentrations of anti-PLA2R-IgG antibodies at initial diagnosis were 5.6 times

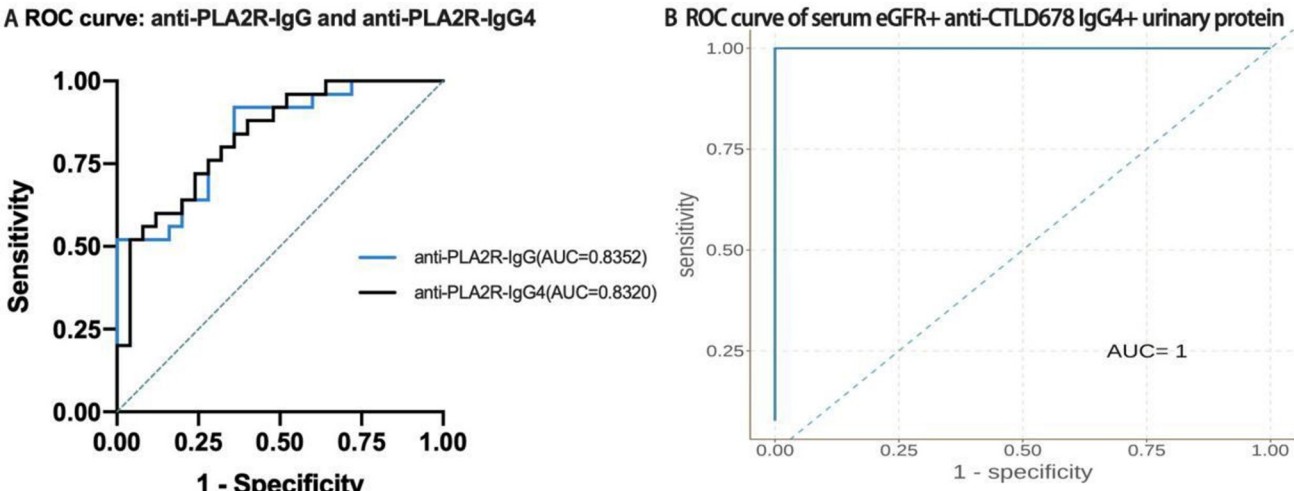

**Fig 4. ROC analysis of serum anti-PLA2R antibodies between the remission and non-remission groups at initial diagnosis.** (A) ROC curve of serum anti-PLA2R-IgG and anti-PLA2R-IgG4 between the remission and non-remission groups. (B)ROC curve of serum eGFR+ anti-CTLD678 IgG4+ urinary protein between the remission and non-remission groups.

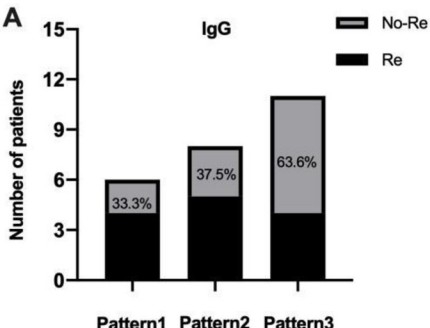
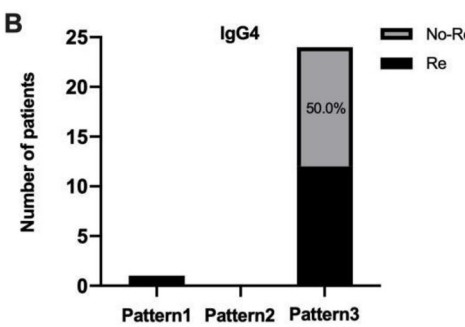

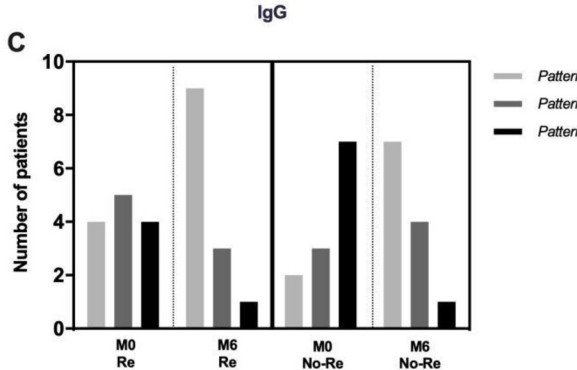
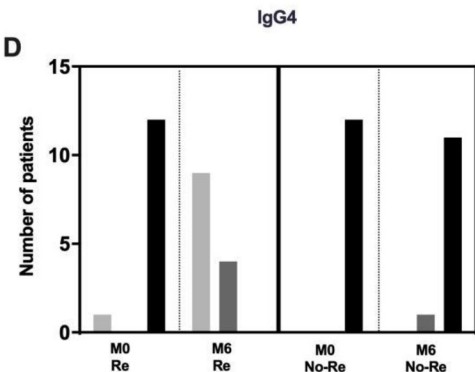

**Fig 5. Epitope characteristics of the remission group and the non-remission group before and after treatment (M0 and M6).**
(A) Detection of numbers of epitopes recognized in the remission and non-remission groups using specific IgG. (B) Detection of numbers of epitopes recognized in the remission and non-remission groups using specific IgG4. (C) Use of IgG-specific to detect epitope characteristics of the remission and non-remission groups before and after treatment (M0 and M6). (D) Use of IgG4-specific to detect epitope characteristics of the remission and non-remission groups before and after treatment (M0 and M6).

higher in patients from the non-remission group than in the remission group, and the difference in antibody concentrations between the two groups at the end of follow-up was also 5.6 times higher. Compared with the patients from the remission group, those from the non-remission group had significantly higher concentrations of anti-PLA2R-IgG4 and domain-specific IgG4 antibodies against CysR and CTLD1 at initial diagnosis. The median concentrations of anti-PLA2R-IgG4 antibody in the patients from the non-remission group were 6.2 times higher than those in patients from the remission group at initial diagnosis. After 6 months of treatment with immunosuppressive drugs, the antibody concentrations were reduced, but the antibody concentration in the remission group was 85.2 times that of the non-remission group at the end of follow-up. This result indicated that anti-PLA2R-IgG4 was more sensitive to treatment than the other antibodies. Further analyses of domain-specific antibody concentrations showed that the median concentrations of anti-CTLD6-7-8-IgG in patients from the non-remission group were 4.2 times higher than those in the remission group, with a 1.5-fold difference in antibody concentrations at the end of follow-up. The median concentrations of anti-CTLD6-7-8-IgG4 antibody in the patients from the non-remission group were 2.6 times higher than those in the remission group at initial diagnosis, but the difference in antibody concentrations between the two groups was 22.3-fold at the end of follow-up. Only slight differences in the concentrations of domain-specific IgG antibodies against CysR and CTLD1 were found between the two groups after follow-up. The results of this study

show that low concentrations of anti-CysR-IgG4, anti-CTLD1-IgG4, and anti-CTLD6-7-8-IgG at initial diagnosis predict rapid remission after treatment, and detection of specific IgG4 for PLA2R and different epitopes can be used to assess therapeutic effects.

The serum concentrations of antibodies against PLA2R and its CysR, CTLD1, and CTLD6-7-8 epitopes were quantitatively detected and compared in patients with IMN at initial diagnosis and the last follow-up (Fig 6). The concentrations of antibodies against PLA2R and its different epitopes decreased in both groups at six months after immunosuppressive therapy relative to those at initial diagnosis. However, the decrease in PLA2R-specific antibody concentrations different between the two groups. Compared with the specific IgG assay, the specific IgG4 assay showed a significant difference in the decrease in antibody concentrations between the two groups.

## 4. Discussion

PLA2R is the principal target antigen of IMN. This protein is a large transmembrane glycoprotein expressed by podocytes. Its expression induces a humoral immune response largely consisting of IgG, which thus generates circulating autoantibodies. IMN can be clinically diagnosed and monitored by testing for PLA2R [12]. Among them, PLA2R-IgG4 antibody is considered to be the most valuable for stratifying the risk of IMN [13]. In recent years, the number of studies focusing on PLA2R has gradually increased. Kao et al. [14] established a series of truncated PLA2R fragments containing different epitopes. A western blot analysis of serum samples containing anti-PLA2R antibody was carried out to confirm that the dominant epitope CysR-FNII-CTLD1, which plays a major role in the structure of PLA2R, is the initial site that can specifically bind to its own anti-PLA2R antibody. Alberto Mella et al. [3] found that marker of urinary podocytes was associated with PLA2R antibodies in membranous nephropathy.

Seitz-Polski et al. [6] found independent epitopes in CysR, CTLD1, and CTLD7 that can generate reactivity against anti-PLA2R antibodies, and they proposed an "epitope spreading" of PLA2R for the first time. Epitope spreading refers to the recognition of a series of structural sites of antigens by autoantibodies in autoimmune diseases or inflammation and the expansion from an immune-dominant epitope to other epitopes within the molecule.

In addition, Seitz-Polski et al. [15] analyzed the influence of the identified PLA2R epitopes on the clinical activities, especially the prognosis, of the IMN. Patients with anti-PLA2R antibodies targeting only the CysR domain have generally mild symptoms and likely experience spontaneous remission, whereas patients with two or three epitopes targeted by anti-PLA2R antibodies have worse symptoms and prognosis.

Recently, Reinhard et al. [7] have adopted the same PLA2R deletion construction strategy used by Seitz-Polski et al. [15]. In a study cohort of 150 patients newly diagnosed with PLA2R-associated MN, results of western blot analysis revealed epitopes in the C-terminal domains (CTLD8 and CTLD7–8) in 16% and 83% of patients with PLA2R-Ab seropositivity, respectively. In addition, a previous study [16] showed evidence that all patients with IMN have antibodies against the N-terminal (CysR and/or CTLD1) epitopes. Therefore, the authors suggested that anti-PLA2R antibodies can recognize at least two epitopes in the N-terminal and C-terminal regions of PLA2R in all patients at clinical diagnosis. However, albuminuria remission occurred in 133 patients and was not dependent on the domain-recognition profiles, which contradicts the idea of epitope spreading proposed by Seitz-Polski et al.

Huang et al. reported that using highly sensitive TRFIA for the quantitative detection of PLA2R IgG and IgG4 subtypes allows good differential diagnosis of IMN [16–18]. Liu et al. [13] found that PLA2R-IgG4 is a more efficient biomarker than PLA2R-IgG for predicting the

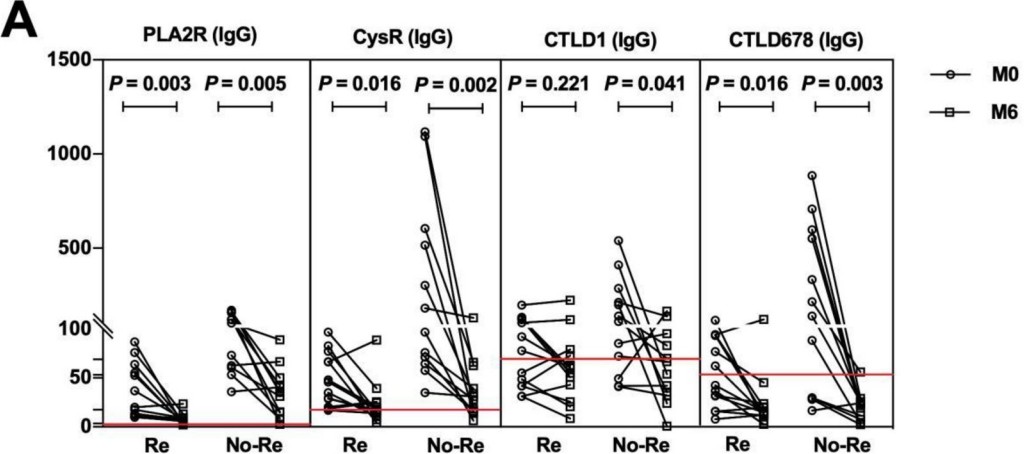

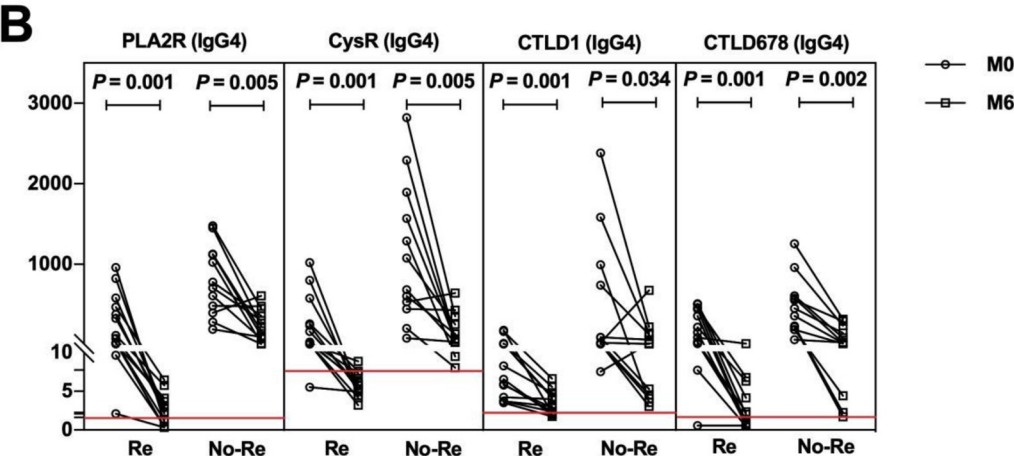

**Fig 6. Changes in concentrations of antibodies against PLA2R and its epitopes before and after treatment.** (A) Concentrations of IgG antibodies against PLA2R and its epitopes in patients with IMN from the remission group (n = 13) and non-remission group (n = 12) at initial diagnosis (M0) and 6 months follow-up (M6). (B) Concentrations of IgG4 antibodies against PLA2R and its epitopes in patients with IMN from the remission group (n = 13) and non-remission group (n = 12) at M0 and M6. "Re" represents the remission group and "No-Re" represents the non-remission group. Red lines represent the cut-off concentrations of PLA2R and its epitopes.

risk of progression in IMN. In the present study, the CysR, CTLD1, and CTLD6-7-8 epitopes of PLA2R were subjected to highly sensitive TRFIA quantification in patients with IMN before and after treatment to determine the clinical value of epitope spreading.

Our study found that the antibodies of most patients with IMN recognized more than two epitopes. After 6 months of treatment, the concentrations of CysR-specific IgG antibodies were greater in many patients from the remission group than in those from the non-remission group, but the concentrations of CTLD1/CTLD6-7-8-specific IgG antibodies showed no significant differences between the two groups. This finding agrees with the report of Reinhard et al. [7]. Thus, the effect of epitope spreading on patient stratification and efficacy is not prominent. In addition, the number of patients whose PLA2R and CysR-, CTLD1-, and CTLD6-7-8-specific IgG4 antibody concentrations became negative was significantly greater in the remission group than in the non-remission group, and the number of identifiable

epitopes was significantly lower in the patients from the remission group than in those from the non-remission group, which supports the findings of Seitz-Polski et al. [15].

Therefore, the simultaneous analyses of specific IgG and IgG4 antibodies against PLA2R and its CysR, CTLD1, and CTLD6-7-8 epitopes through highly sensitive quantitative assays can explain the different views in previous reports. Epitope spreading does not play a significant role in evaluating the therapeutic effect if only the numbers of identifiable epitopes are qualitatively determined by specific IgG antibodies. However, the highly sensitive quantitative detection of changes in the concentrations of specific IgG4 antibodies against PLA2R and its CysR, CTLD1, and CTLD6-7-8 epitopes can be a good prognostic tool. Therefore, a quantitative detection rather than a qualitative analysis of epitope spreading is preferred.

This study has certain limitations. Given that the follow-up results were evaluated after 6 months of immunosuppressive therapy, data at this point indicated that the concentrations of IgG4 antibodies against PLA2R and its epitopes had largely returned to normal in patients from the remission group. However, it is possible that the concentrations of IgG4 antibodies against PLA2R and its epitopes had returned to normal earlier than 6 months, which may be suitable for accurate evaluation.

In conclusion, the combined quantitative detection of specific IgG and IgG4 antibodies against PLA2R and its CysR, CTLD1, and CTLD6-7-8 epitopes can identify epitope spreading in patients with IMN, and the detection of specific IgG4 antibodies can be more used for prognosis. Clinicians can use the concentrations of these antibodies in patients as a reference in deciding the most effective treatment strategy for their patients.

## Supporting information

**S1 Checklist. STROBE statement—checklist of items that should be included in reports of observational studies.**
(DOCX)

**S1 Data.**
(XLSX)

## Acknowledgments

We thank for Ting Li and Professor Biao Huang providing us the highly sensitive time-resolved fluorescence immunoassay to quantitatively detect the concentrations of different antibodies against epitopes of PLA2R.

## Author Contributions

**Conceptualization:** Yuan Qin, Zijian Huang.

**Data curation:** Jing Xue, Qingqing Wu, Bo Lin, Yuan Qin.

**Formal analysis:** Jing Xue, Yuan Qin, Liang Wang.

**Funding acquisition:** Liang Wang.

**Investigation:** Xiaobin Liu, Huiming Sheng, Xue Yang, Zhigang Hu.

**Methodology:** Ting Li, Qingqing Wu, Huiming Sheng, Xue Yang, Zijian Huang, Liang Wang, Zhigang Hu, Biao Huang.

**Project administration:** Huiming Sheng, Bo Lin, Liang Wang.

**Software:** Xiumei Zhou, Leting Zhou.

**Validation:** Xue Yang, Leting Zhou.

**Visualization:** Xiumei Zhou, Leting Zhou.

**Writing – original draft:** Xiaobin Liu, Biao Huang.

**Writing – review & editing:** Jing Xue, Zijian Huang, Leting Zhou, Liang Wang, Biao Huang.

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
