## [Decision Letter · Decision Letter 0]

24 Oct 2023

PONE-D-23-27578Quantitative Detection and Prognostic Value of Antibodies Against M-type Phospholipase A2 Receptor and its Cysteine-Rich Ricin Domain and C-type Lectin Domains 1 and 6-7-8 in Patients with Idiopathic Membranous NephropathyPLOS ONE

Dear Dr. Wang,

Thank you for submitting your manuscript to PLOS ONE. After careful consideration, we feel that it has merit but does not fully meet PLOS ONE’s publication criteria as it currently stands. Therefore, we invite you to submit a revised version of the manuscript that addresses the points raised during the review process.

We look forward to receiving your revised manuscript.

Kind regards,

Joseph J Barchi

Academic Editor

PLOS ONE

Journal Requirements:

"This study was supported by the Social Development Fund of Zhejiang Province [No. LGF20H200008]; Chinese National Natural Science Foundation [No. 82172336,81672083,82070730]; Key Research and Development Project of Zhejiang Province (No. 2022C03118), the Natural Science Foundation of Zhejiang Province(LQ23H050005), Key Research and Development Project of Hangzhou (No. 202004A23),Precision medicine key project of Wuxi Health Commission [No. J202001], theTop Talent Support Program for young and middle-aged people of Wuxi Health Committee  [HB2020008],Wuxi Medical Innovation Team Project [CXTD2021010],Specialized Disease Cohort of Wuxi Medical Center of Nanjing Medical University(WMCC202316)."

7. Your ethics statement should only appear in the Methods section of your manuscript. If your ethics statement is written in any section besides the Methods, please move it to the Methods section and delete it from any other section. Please ensure that your ethics statement is included in your manuscript, as the ethics statement entered into the online submission form will not be published alongside your manuscript.

**Additional Editor Comments:**

Please pay attention to both reviews and respond accordingly. I would be happy to consider a revised manuscript if all reviewer questions are answered.

Reviewers' comments:

Reviewer's Responses to Questions

**Comments to the Author**

1. Is the manuscript technically sound, and do the data support the conclusions?

Reviewer #1: Yes

Reviewer #2: Partly

2. Has the statistical analysis been performed appropriately and rigorously? 

Reviewer #1: Yes

Reviewer #2: Yes

3. Have the authors made all data underlying the findings in their manuscript fully available?

Reviewer #1: Yes

Reviewer #2: Yes

4. Is the manuscript presented in an intelligible fashion and written in standard English?

Reviewer #1: Yes

Reviewer #2: Yes

5. Review Comments to the Author

Reviewer #1: The study aimed to evaluate the clinical value of epitope spreading in patients with idiopathic membranous nephropathy (IMN) by quantitatively detecting changes in the concentrations of different antibodies against epitopes of the M-type phospholipase A2 receptor (PLA2R) before and after treatment. The results showed that the concentration of specific IgG and IgG4 antibodies against PLA2R and its epitopes was higher in the non-remission group compared to the remission group. Low concentrations of anti-CysR-IgG4, anti-CTLD1-IgG4, and anti-CTLD6-7-8-IgG4 at the initial diagnosis predicted rapid remission after treatment. The use of specific IgG4 against PLA2R and its different epitopes combined with estimated glomerular filtration rate (eGFR) and urinary protein provided a better assessment of the prognostic outcome of IMN.

As a nephrologist reviewing this abstract, I have several questions:

1. CYC regimen was administered to the most patients in both groups, while fewer patients received the CNI regimen or rituximab. Is there any selection bias? If so, did this impact the antibody responses?

2. How representative is the patient cohort in this study of the broader population of IMN patients? Are there any specific inclusion or exclusion criteria that could limit the generalizability of these findings to other populations?

3. In the literature (PMCID: PMC4849812), epitope spreading during follow-up is associated with disease worsening, whereas reverse spreading from a CysRC1C7 profile back to a CysR profile is associated with a favorable outcome. I wonder whether the author of this study found similar findings in the Chinese cohort of membranous nephropathy.

4. Were long-term outcomes, such as the risk of relapse or the need for further interventions, assessed in this study? It would be interesting to know whether the initial antibody concentrations and epitope-specific responses are predictive of sustained remission.

Reviewer #2: The article is well structured, with a clear abstract summarizing the study's object. In my opinion, some points need to be addressed and limit the generalizability of the study.

- Limited sample size, which strictly difference every conclusion about AUC

- Selection criteria (all patients underwent different lines of treatment, with different approaches potentially related to differential effects on Ab anti-PLA2R)

- As reported in KDIGO guidelines, monitoring antibodies against different antigens of PLA2R can be considered a good proposal for immunological monitoring after therapy in patients with membranous glomerulonephritis. However, the innovative contribution of this publication in the management of GNM appears to be limited (KDIGO 2021)

- Additional markers could be tested to improve the significance of epitope spreading (i.e., podocyturia, see in example 10.1038/s41598-020-73335-2). Please comment on these data in the introduction and discussion

6. PLOS authors have the option to publish the peer review history of their article (what does this mean?). If published, this will include your full peer review and any attached files.

Reviewer #1: No

Reviewer #2: No

---

## [Author Response · Author response to Decision Letter 0]

16 Dec 2023

Dear Editor and Reviewers,

Thank you for your letter and for the reviewers’ comments concerning our manuscript entitled

’Quantitative Detection and Prognostic Value of Antibodies Against M-type Phospholipase A2 Receptor and its Cysteine-Rich Ricin Domain and C-type Lectin Domains 1 and 6-7-8 in Patients with Idiopathic Membranous Nephropathy’(PONE-D-23-27578). Those comments are all valuable and very helpful for revising and improving our paper. We have studied comments carefully and have made correction which we hope meet with approval. The reviewers’ comments are laid out below in italicized font and specific concerns have been numbered.Our response is given in normal font and changes to the manuscript are given in the red text.

Replies to the reviewers’ comments:

Reviewer #1:Reviewer #1: The study aimed to evaluate the clinical value of epitope spreading in patients with idiopathic membranous nephropathy (IMN) by quantitatively detecting changes in the concentrations of different antibodies against epitopes of the M-type phospholipase A2 receptor (PLA2R) before and after treatment. The results showed that the concentration of specific IgG and IgG4 antibodies against PLA2R and its epitopes was higher in the non-remission group compared to the remission group. Low concentrations of anti-CysR-IgG4, anti-CTLD1-IgG4, and anti-CTLD6-7-8-IgG4 at the initial diagnosis predicted rapid remission after treatment. The use of specific IgG4 against PLA2R and its different epitopes combined with estimated glomerular filtration rate (eGFR) and urinary protein provided a better assessment of the prognostic outcome of IMN.

The author’s response:We are really grateful to Review#1 for his/her effort reviewing our paper and his/her positive feedback.The summary of our work as written by this reviewer is precise.

As a nephrologist reviewing this abstract, I have several questions:

1. CYC regimen was administered to the most patients in both groups, while fewer patients received the CNI regimen or rituximab. Is there any selection bias? If so, did this impact the antibody responses?

The author’s answer: Thank you for your careful reading. This study is a small sample observational study, including all patients admitted to the Department of Nephrology, Wuxi People's Hospital Affiliated to Nanjing Medical University March 2016 to December 2016 who were diagnosed with primary membranous nephropathy by renal biopsy. The immunosuppressive treatment regimen was selected in accordance with the risk stratification of KDIGO guidelines and the wishes of patients and/or their families, regardless of the subjective intent of the investigator. The correlation between PLA2R antibody level and prognosis of patients with nephropathy was not affected by immune intervention regimen. Therefore, these patients with membranous nephropathy were studied as a whole, without differentiating among the immunological interventions. Follow-up studies will be conducted to minimize selection bias by expanding the sample size.

2. How representative is the patient cohort in this study of the broader population of IMN patients? Are there any specific inclusion or exclusion criteria that could limit the generalizability of these findings to other populations?

The author’s answer: Thank you for your question. As mentioned above, our study is a single-center small sample observational study. In this study, all adult patients diagnosed with membranous nephropathy by renal biopsy in the Department of Nephrology of our hospital March 2016 to December 2016 were selected and followed up for 6 months. Secondary membranous nephropathy, concomitant infections and loss of follow-up were excluded.

3. In the literature (PMCID: PMC4849812), epitope spreading during follow-up is associated with disease worsening, whereas reverse spreading from a CysRC1C7 profile back to a CysR profile is associated with a favorable outcome. I wonder whether the author of this study found similar findings in the Chinese cohort of membranous nephropathy.

The author’s answer: Similar to the results of Seitz-Polsk et al., our study found that after six months of treatment, the number of epitopes in the remission group was significantly reduced compared with that before treatment (see Figure 5). However, limited to a small sample study, no difference was found in PLA2R antibody domain epitope spreading before and after treatment between the remission group and the non-remission group. We will expand the sample size to further validate the association between PLA2R epitope spreading and remission of membranous nephropathy.

4. Were long-term outcomes, such as the risk of relapse or the need for further interventions, assessed in this study? It would be interesting to know whether the initial antibody concentrations and epitope-specific responses are predictive of sustained remission.

The author’s answer: Thank you for your constructive comments. We will extend the observation period to calculate the recurrence of these patients and further immunological intervention within 5 years, and further investigate the relationship between PLA2R antibody and epitope spread and the risk of recurrence or further immune intervention in these patients to understand whether initial antibody concentrations and epitope-specific responses predict sustained remission.

Reviewer #2:  The article is well structured, with a clear abstract summarizing the study's object. In my opinion, some points need to be addressed and limit the generalizability of the study.

1.Limited sample size, which strictly difference every conclusion about AUC

The author’s answer: Limited to a single-center small sample observational study, we initially explored the relationship between PLA2R antibody and epitope and the prediction of remission of membranous nephropathy. We will conduct further research on the results of the current observation with a large sample of multi-centers.

2.Selection criteria (all patients underwent different lines of treatment, with different approaches potentially related to differential effects on Ab anti-PLA2R)

The author’s answer: Thank you for careful reading.Immunosuppressive regimen may affect serum PLA2R antibody levels and response rate of membranous nephropathy at follow-up.The objective of this study was the prediction of remission of membranous nephropathy by serum PLA2R antibody and its domains, so we studied these patients as a group and did not distinguish between the immunosuppressants patients were receiving.Subsequently, we plan to expand the sample size and conduct a group study of patients receiving different immune interventions to explore the effect of immunosuppressors on PLA2R antibody and renal remission.

3. As reported in KDIGO guidelines, monitoring antibodies against different antigens of PLA2R can be considered a good proposal for immunological monitoring after therapy in patients with membranous glomerulonephritis. However, the innovative contribution of this publication in the management of GNM appears to be limited (KDIGO 2021)

The author’s answer: Thank you for your constructive comments. The 2021 KDIGO Guidelines recommend PLA2R-IgG antibodies for the diagnosis, risk stratification, and prognosis of PMN. However, in clinical work, we have found PLA2R-negative patients with membranous nephropathy and patients with poor response to immune interventions, so PLA2R antibodies cannot accurately monitor the diagnosis and treatment of all patients with membranous nephropathy. We further investigated the relationship between PLA2R antibody specificity and the remission of membranous nephropathy proteinuria at antibody subgroup level and antigen epitope level, in order to better manage PLA2R-associated membranous nephropathy patients.

4. Additional markers could be tested to improve the significance of epitope spreading (i.e., podocyturia, see in example 10.1038/s41598-020-73335-2). Please comment on these data in the introduction and discussion

The author’s answer: Thank you for your constructive suggestions to improve the quality of the discussion. This study suggests that podocyturia correlate with PLA2R antibody levels, providing a new predictor of disease severity in patients with membranous nephropathy. I cited this study in my introduction and discussion.

We appreciate for the editor/reviewers' warm work earnestly, and hope that the correction will meet with approval.

Once again, thank you very much for your comments and suggestions.

Kind regards.

Jing Xue

E-mail:475809491@qq.com

Corresponding author: Liang Wang

E-mail:wangliang_wuxi@126.com

---

## [Decision Letter · Decision Letter 1]

23 Jan 2024

Quantitative Detection and Prognostic Value of Antibodies Against M-type Phospholipase A2 Receptor and its Cysteine-Rich Ricin Domain and C-type Lectin Domains 1 and 6-7-8 in Patients with Idiopathic Membranous Nephropathy

PONE-D-23-27578R1

Dear Dr. Wang,

We’re pleased to inform you that your manuscript has been judged scientifically suitable for publication and will be formally accepted for publication once it meets all outstanding technical requirements.

Kind regards,

Joseph J Barchi

Academic Editor

PLOS ONE

Additional Editor Comments (optional):

Reviewers' comments:

Reviewer's Responses to Questions

**Comments to the Author**

1. If the authors have adequately addressed your comments raised in a previous round of review and you feel that this manuscript is now acceptable for publication, you may indicate that here to bypass the “Comments to the Author” section, enter your conflict of interest statement in the “Confidential to Editor” section, and submit your "Accept" recommendation.

Reviewer #2: All comments have been addressed

2. Is the manuscript technically sound, and do the data support the conclusions?

Reviewer #2: Yes

3. Has the statistical analysis been performed appropriately and rigorously? 

Reviewer #2: Yes

4. Have the authors made all data underlying the findings in their manuscript fully available?

Reviewer #2: Yes

5. Is the manuscript presented in an intelligible fashion and written in standard English?

Reviewer #2: Yes

6. Review Comments to the Author

Reviewer #2: All suggested comments have been appropriately addressed. The quality of the paper is now improved. I have no further comments.

7. PLOS authors have the option to publish the peer review history of their article (what does this mean?). If published, this will include your full peer review and any attached files.

Reviewer #2: No

---

## [Editor Report · Acceptance letter]

13 Feb 2024

PONE-D-23-27578R1 

PLOS ONE

Dear Dr. Wang, 

I'm pleased to inform you that your manuscript has been deemed suitable for publication in PLOS ONE. Congratulations! Your manuscript is now being handed over to our production team.

Kind regards, 

on behalf of

Dr. Joseph J Barchi 

Academic Editor

PLOS ONE